# The Unknown Health Burden of Herpes Zoster Hospitalizations: The Effect on Chronic Disease Course in Adult Patients ≥50 Years

**DOI:** 10.3390/vaccines8010020

**Published:** 2020-01-10

**Authors:** Maria Francesca Piazza, Chiara Paganino, Daniela Amicizia, Cecilia Trucchi, Andrea Orsi, Matteo Astengo, Paolo Romairone, Simona Simonetti, Giancarlo Icardi, Filippo Ansaldi

**Affiliations:** 1Department of Health Sciences, University of Genoa, Via Antonio Pastore 1, 16132 Genoa, Italy; mariafrancesca.piazza@regione.liguria.it (M.F.P.); daniela.amicizia@unige.it (D.A.); andrea.orsi@unige.it (A.O.); matteoastengo@gmail.com (M.A.); icardi@unige.it (G.I.); filippo.ansaldi@unige.it (F.A.); 2A.Li.Sa., Liguria Health Authority, Piazza della Vittoria 15, 16121 Genoa, Italy; cecilia.trucchi@regione.liguria.it; 3Hygiene Unit, San Martino Polyclinic Hospital, Largo Rosanna Benzi 10, 16132 Genoa, Italy; 4Liguria Digitale S.p.A., Via Enrico Melen 77, 16152 Genoa, Italy; p.romairone@liguriadigitale.it (P.R.); S.simonetti@liguriadigitale.it (S.S.)

**Keywords:** Herpes Zoster, hospitalizations, chronic diseases, risk factors, high-risk groups, elderly, cerebral vasculopathy, readmissions and drug therapy

## Abstract

The effect of severe Herpes Zoster (HZ) on chronic diseases is a component of the real burden of this vaccine-preventable disease that is not commonly considered. A retrospective cohort study was conducted to assess the health burden of severe HZ in adults ≥50 years residing in Liguria Region from 2015 to 2017. Subjects hospitalized with and without HZ were matched (1:6 ratio). 437 subjects in the HZ cohort and 2622 subjects in the non-HZ cohort were enrolled. Previous immunodeficiency, autoimmune, and rare diseases are identified as main chronic conditions related to HZ hospitalization. Higher incidences of autoimmune (1.4% vs. 0.22%, *p* = 0.002) and gastrointestinal (7.04% vs. 3.62%, *p* = 0.015) diseases after hospitalization were observed in the HZ cohort compared to the non-HZ cohort. Significantly higher incidences were found after hospitalization versus the previous period for cardiovascular diseases (11.17% vs. 2.09%, *p* < 0.001), cerebral vasculopathy (6.13% vs. 0.60%, *p* < 0.001), non-arrhythmic myocardiopathy (4.31% vs. 0.59%, *p* = 0.002), and neuropathy (2.62% vs. 0.56%, *p* = 0.033). The HZ cohort showed a relative risk 10-fold higher for cerebral vasculopathy, 5-fold higher for cardiovascular diseases, and 7-fold higher for non-arrhythmic myocardiopathy. HZ causes a substantial impact on the chronic conditions. These data could suggest an implementation of HZ vaccination programs in the elderly and in high-risk groups.

## 1. Introduction

Herpes Zoster (HZ) is the clinical manifestation of reactivation of varicella-zoster virus remained latent in dorsal root or cranial nerve sensory ganglia after primary infection. It is a considerable cause of morbidity, especially in elderly, immunosuppressed, or critically ill patients [1]. In Europe, approximately 1.7 million new cases of HZ are expected each year with an incidence rate in adults aged 50 years and older, between 6 and 8 cases per 1000 person-years [2,3,4,5].

The incidence shows a growing trend with increasing age until reaching over 10 cases per 1000 person-years after 80 years old [3,5,6]. The most common complications related to HZ are post-herpetic neuralgia (PHN), a long-lasting pain that does not end with the disappearance of the rash, but persists for some weeks, and even months or years; ocular lesions involving the ophthalmic division of the trigeminal nerve; neurological sequelae; and visceral inflammation [5,7,8,9]. Severe cases with these complications often require hospitalization [6,10,11].

The health and economic burden of HZ in terms of incidence and neurological complications—PHN, ocular HZ, etc.—and its costs were deeply investigated, as well as risk factors and chronic diseases (i.e., chronic renal failure, chronic obstructive pulmonary disease (COPD), diabetes, asthma, and depression) associated with HZ onset [12,13,14,15]. The effect of severe HZ on the clinical course of chronic diseases was poorly explored in terms of new cases due to clinical onset of latent disease, worsening of course, hospitalizations, and increase of medication use.

A retrospective cohort study was carried out to assess the health effects of severe HZ requiring hospitalization on clinical progression of chronic diseases in Ligurian residents aged 50 years or older. Prevalence of chronic diseases at 6 months before hospitalization and incidence observed during the 6 months after hospitalization were compared in patients hospitalized for HZ and in an age- and gender-matched non-HZ cohort. Furthermore, incidence rates in the 6 month periods before and after hospitalization were compared in patients who accessed the hospital for HZ.

## 2. Materials and Methods

### 2.1. Study Design and Patients Selection Criteria

This retrospective cohort study was conducted to assess the epidemiological health burden of severe HZ in adults ≥50 years residing in the Liguria Region from January 2015 to December 2017. Subjects were included in the HZ cohort on the basis of International Classification of Diseases 9th Revision, Clinical Modification (ICD-9-CM) diagnosis and procedure codes selecting patients who were hospitalized with a primary or secondary diagnosis of HZ (ICD-9-CM codes 053.xx) [16,17]. The HZ cohort subjects were matched to hospitalized patients without a diagnosis of HZ infection (non-HZ cohort) according to age, gender, and admission date in a 1:6 ratio. The subjects in the non-HZ cohort were enrolled at the same time as each matched HZ patient (±15 days) by random selection. The subjects of the HZ cohort and non-HZ cohort were randomly selected among all hospitalized patients resident in Liguria who had access to a hospital of the National Health Service. In Italy, hospital access is free of charge. All patients were monitored in the 6 months preceding hospital admission (pre-H) and following the discharge (post-H) in order to estimate the prevalence and incidence of chronic diseases. Chronic disease cases captured during hospitalization were excluded from the analysis. Days of drug therapy and further hospitalizations were also compared during pre-H and post-H periods. Drugs were classified using the Anatomical Therapeutic Chemical (ATC) international classification system.

### 2.2. Data Sources and Information Flows

Administrative healthcare data (AHD), or *data warehouse,* is a regional service that collects Hospital Discharge Records (HDRs), the flow of outpatient visits, and pharmaceutical and was used as a data source. The HDRs include patient demographic data, admission and discharge dates, discharge status, main and secondary discharge diagnoses, and diagnostic/therapeutic procedures. HDR data are recorded with the consent of the patient and can be used for scientific studies in the form of aggregated and de-identified data. Data about the chronic diseases were obtained through the Liguria Chronic Condition Data Warehouse (CCDWH) platform, recorded in the Liguria region since 2010 [18].

### 2.3. Data Analysis

Data were analyzed by JMP version 13.0.0 software (SAS Institute, Cary, NC, USA). Continuous variables were summarized as medians and interquartile range, and categorical variables as frequency distributions and 95% confidence intervals. Prevalence, incidence, and risk measures, i.e., odds ratio and relative risk, were reported with a 95% confidence interval.

The chi-square test (Pearson Test) and Mann–Whitney U test were used to compare the general features, comorbidities, readmissions, and days of drug therapy in HZ cohort and non-HZ cohort groups. Comparison of variables at the 6 months before and after hospitalization in the HZ cohort and non-HZ cohort groups was carried out using the Mann–Whitney test when appropriate. Statistical significance was defined as *p*-value < 0.05.

### 2.4. Ethical Statement

Data collected in the AHD regional service are transmitted by the Ligurian Local Health Units (LHUs) and hospitals to the regional health authority “Azienda Ligure Sanitaria” (A.Li.Sa.). The institutional activities of A.Li.Sa. include handling regional healthcare administrative data and conduct epidemiological studies, projects and researches to support strategical choices of healthcare [19]. The current regulation on privacy allows professionals belonging to the regional A.Li.Sa. accessing to healthcare administrative data routinely transmitted by the public funded LHUs and hospitals of the Liguria region. Data are anonymous and have been evaluated in an aggregate manner. Finally, patients who access to regional public health services give consent to the use of healthcare data also for scientific purpose.

## 3. Results

The study population included 437 hospitalized subjects with a primary and/or secondary diagnosis of HZ and 2622 subjects without an HZ diagnosis. The study sample was recruited from the period of 2015–2017. The overall HZ hospitalization rate amounted to 18.16 cases per 100,000 residents. Data stratified by age group and according to admission year showed an increase in hospitalizations with age, reaching rates >50 cases per 100,000 residents in adults aged 84 years and older (Figure 1).

Median age at hospitalization date was 80 years (IQR: 71–86 years) and women represented 54% of patients in the HZ and non-HZ cohorts; length of hospital stay was significantly higher in the HZ cohort (median: 9 days; IQR: 6–17 days) in comparison with the non-HZ cohort (median: 7 days; IQR: 3–16 days, *p*-value < 0.001).

In the HZ-cohort, 49.7% of cases reported a primary and/or secondary diagnosis of HZ without complications (ICD-9-CM code 053.9), and only 2% were enrolled with a primary diagnosis of postherpetic neuralgia/polyneuropathy (ICD-9-CM codes: 053.12, 053.13).

The prevalence of chronic diseases at 6 months before hospitalization in the HZ and non-HZ cohorts is reported in Table 1.

The HZ cohort showed a significant higher prevalence of patients with previous transplantation (1.83% vs. 0.34%; *p* < 0.001), HIV positivity (1.37% vs. 0.04%; *p* < 0.001), autoimmune diseases (4.12% vs. 2.33%; *p* = 0.029) and rare diseases (1.14% vs. 0.31%, *p* = 0.013) compared to the non-HZ cohort. Autoimmune diseases were detected in 18 subjects in the HZ cohort including rheumatoid arthritis (10 cases), myasthenia gravis, systemic lupus erythematosus, Sjogren’s syndrome, Hashimoto’s thyroiditis, systemic sclerosis, and autoimmune hemolytic anemia. Rare diseases observed in five subjects belonging to the HZ cohort included (i) diseases of the endocrine glands, nutrition, metabolism, and immune disorders; (ii) blood and hematopoietic organ diseases, and (iii) diseases of the digestive system for which it is approved a specific case definition for rare diseases by the Italian Ministry of Health (DPCM—Prime Ministerial Decree of 12 January 2017).

Incidence rates of chronic diseases during the 6 months after hospitalization in the HZ cohort and non-HZ cohort are shown in Table 2.

Significant higher incidence rates of autoimmune (1.4% vs. 0.22%, *p* = 0.002) and gastrointestinal (7.04% vs. 3.62%, *p* = 0.015) diseases during the 6 months after hospitalization were observed in the HZ cohort compared to the non-HZ cohort group. Among gastrointestinal diseases, the most relevant differences between the two cohorts regard chronic hepatitis, liver cirrhosis, and gastroesophageal reflux disease (GERD). The incidence rate of diabetes was twice as high in the HZ cohort as it was in the non-HZ group, and the difference resulted at the limit of statistical significance (2.61% vs. 1.16%, *p* = 0.054).

In Table 3 are represented the cumulative incidence rates of chronic diseases in the follow-up periods before and after hospitalization for the HZ cohort group.

Significantly higher incidence rates were found in the 6 months after hospitalization versus the 6 months before for cardiovascular diseases, excluding subjects with only hypertension (11.17% = *n*. 23 vs. 2.09% = *n*. 5, *p* < 0.001), cerebral vasculopathy including acute stroke and transient ischemia (6.13% = *n*. 19 vs. 0.60% = *n*. 2, *p* < 0.001), non-arrhythmic myocardiopathy (4.31% = *n*. 14 vs. 0.59% = *n*. 2, *p* = 0.002), and neuropathy (2.62% = *n*. 9 vs. 0.56% = *n*. 2, *p* = 0.033) including neurodegenerative conditions and epilepsies. In particular, the patients with neuropathy in the 6 months after hospitalization were affected in most cases (55.6%) by dementia, showing a median age of 87 years old. In these patients, dementia was associated with cerebral vasculopathy in 60% of cases.

Furthermore, Table 3 shows higher incidence rates at the margin of statistical significance for not complicated diabetes (2.52% vs. 0.61%, *p* = 0.059) and gastrointestinal diseases (7.04% vs. 3.41%, *p* = 0.062) including GERD (5.54% vs. 2.33%, *p* = 0.052) in the HZ cohort.

Readmission within 6 months from hospitalization was analyzed, showing a rehospitalization rate of 46.3% in the HZ cohort and 50.1% in the non-HZ cohort with no statistically significant differences between the two groups (*p* = 0.174).

A higher number of further hospitalizations was observed in the HZ cohort than the matched non-HZ cohort group in the 6 months after the admission date for the following major diagnostic categories (MDCs): MDC2—diseases and disorders of the eye (1.59% vs. 0.54%; *p* = 0.031)—and MDC18—infectious and parasitic diseases (systemic or of unspecified sites) (3.97% vs. 2.01%; *p* = 0.026) (Appendix A). In particular, 15 patients had readmissions for MDC18, of which only 3 were related to HZ with specified or unspecified complications (ICD-9-CM diagnosis codes 053.79 and 053.10). Other infectious diagnoses were represented by septicemia, bacterial or viral infections, and kidney or urinary tract infections.

The difference in further hospitalization during the post-H and pre-H periods was also compared: a higher number of further hospitalizations for MDC2 (*p* = 0.023) and MDC18 (*p* = 0.017) was recorded in the HZ cohort compared to the non-HZ cohort in the 6 months after hospitalization.

Statistically significant differences in days of drug therapy between HZ and non-HZ cohorts were observed in the 6 months following hospitalization. In particular, patients with HZ had a higher number of days of drug therapy for anti-infectives for systemic use (ATC-J median: 15, IQR: 7–29 vs. median: 12, IQR: 6–22, *p* = 0.001) and opioids for systemic use (ATC-N02A median: 16, IQR 6–52 vs. 11, IQR 5–38, *p* < 0.001) (Appendix A).

Comparing the 6 months before and after hospitalization date, a longer period of treatment was observed for the HZ cohort compared with the non-HZ cohort for antiviral drugs for systemic use (ATC-J05 *p* = 0.004), including drugs potentially used for HZ therapy, such as valaciclovir (ATC-J05AB11 *p* < 0.001), famciclovir (ATC-J05AB09 *p* = 0.005), acyclovir (ATC-J05AB01 *p* = 0.029), and brivudine (ATC-J05AB15 *p* < 0.001). Days of drug therapy used for treatment of nervous system diseases were statistically higher for the HZ cohort (ATC-N *p* < 0.001), including drugs used in the PHN treatment, such as pregabalin (ATC-N03AX16 *p* < 0.001) and gabapentin (ATC-N03AX12 *p* < 0.001). In addition, the days of drug therapy used in the treatment of pain, such as amitriptyline (ATC-N06AA09 *p* < 0.001), topical lidocaine (ATC-N01BB02 *p* < 0.001), and opioids for systemic use (ATC-N02A *p* = 0.003), showed higher values in the HZ cohort compared with the non-HZ cohort in the 6 months following the hospitalization compared to the previous 6 months. No significant differences resulted in days of therapy related to diabetes treatment. Although the economic burden analysis is not the objective of this study, it is noteworthy that in the HZ cohort 6 months after hospitalization, an incremental cost for diabetes drugs was revealed (+18.56%).

## 4. Discussion

Acute diseases may determine a negative impact on clinical course of chronic diseases because they may worsen chronic cardiovascular, respiratory, liver, or kidney diseases or may challenge individual ability to take prescribed medication. The effect of severe HZ on chronic diseases and comorbidities in terms of new cases, worsening of course, hospitalizations, and increase of medication use is a component of the real burden of this vaccine-preventable disease that is not commonly considered.

This retrospective cohort study assessed the health effects of severe HZ requiring hospitalization in adults aged 50 years or older. Comparing incidence of chronic diseases observed 6 months after and before the hospitalization for HZ, higher relative risks of diagnosis of cardiovascular diseases, cerebral vasculopathy (including acute stroke and transient ischemia), non-arrhythmic myocardiopathy, and neuropathy were identified. Furthermore, the higher number of new cases of not complicated diabetes and gastrointestinal diseases (including GERD) is confirmed in the HZ cohort after hospitalization compared to the previous period, although with incidence difference at the limit of statistical significance.

Recently, the association between cerebrovascular events after an episode of HZ, probably triggered by the replication of varicella-zoster virus at the level of cerebral arterial walls, has been studied in the scientific literature. The infection would develop along the nerve fibers up to the arterial vessels, inducing inflammation and thrombosis with the consequent onset of cerebral vasculopathy [20,21,22,23]. A systematic review and meta-analysis carried out by Zhang et al. in 2017 [20] explored this association, and HZ infection was significantly associated with a slightly increased risk of acute stroke/transient ischemia (RR: 1.30, 95% CI: 1.17–1.46) or myocardial infarction (RR: 1.18, 95% CI: 1.07–1.30). Our results showed a 10-fold higher risk (RR: 10.3, 95% CI: 2.78–65.47) to develop a cerebral vasculopathy for patients hospitalized for HZ, and a high proportion of new cases of cerebral vasculopathy was associated with the onset of dementia.

Furthermore, the study showed a fivefold higher risk (RR: 5.34, 95% CI: 2.13–15.77) to develop cardiovascular diseases excluding hypertension, and in particular, a sevenfold higher risk (RR: 7.26, 95% CI: 1.88–47.16) to develop non-arrhythmic myocardiopathy in the HZ cohort.

Moreover, a higher incidence of diabetes and autoimmune and gastrointestinal diseases (in particular, chronic hepatitis, liver cirrhosis, and GERD) observed in the HZ cohort in comparison with the non-HZ cohort after hospitalization suggests that severe HZ may itself be a risk factor for an acute exacerbation of chronic diseases.

Previous immunodeficiency conditions (transplantation and HIV positive), autoimmune diseases, and rare diseases are confirmed as main chronic conditions related to HZ hospital admission, as other studies reported [24,25,26,27,28,29,30,31]. It was observed that HZ could be a cause of more severe clinical course in immunocompromised patients, particularly in those with decrease of cell-mediated immune response. [32,33].

A higher number of further hospitalizations for infectious and parasitic diseases (MDC 18) was recorded in the HZ cohort compared to the non-HZ cohort in the 6 months after hospitalization, and few cases were related to HZ complications. These data suggest that HZ could be a predisposing factor for the occurrence of other infectious diseases, such as septicemia, bacterial or viral infections, and kidney or urinary tract infections.

Relative to drug consumption, our findings confirm the expected increase of antiviral drugs used in the HZ treatment, drugs used in the postherpetic neuralgia, and drugs used in the pain treatment in HZ cohort [34,35,36].

A strength of the present study is that we investigated HZ-related hospitalizations through data collected from all over the Liguria Region (population 1.56 million). The CCDWH platform consists in a sophisticated system that integrates several administrative regional flows and is used to identify comorbidities with accuracy and precision.

Certain limitations of the study are that our data based on a large administrative database may pose concerns relating to miscoding and data reporting inaccuracies. Furthermore, this study reports only severe cases requiring hospitalization, not considering mild cases or HZ in less frail patients.

Worsening of the clinical picture, which is related to many chronic diseases following HZ hospitalization, has not been observed for some chronic diseases such as renal failure and bronchopneumopathy including asthma. The lack of difference between before and after HZ hospitalization could be due to potentially low sensitivity of the CCDWH system in identifying some minimal clinical changes. This aspect could be explored in further analysis.

Finally, days of drug therapy based on prescriptions may be underestimated because medications paid entirely out of pocket by the subjects were not included in the regional pharmaceutical flow.

A demographic and epidemiological picture characterized by a progressive increase in frail and elderly subjects, an increase in HZ cases, and associated complications, is expected in the next future.

## 5. Conclusions

Our study demonstrates that HZ causes a substantial impact on chronic conditions, better defining the picture of the real burden of this vaccine-preventable disease. These data support implementation of HZ vaccination programs in the elderly and in high-risk groups.

## Figures and Tables

**Figure 1 vaccines-08-00020-f001:**
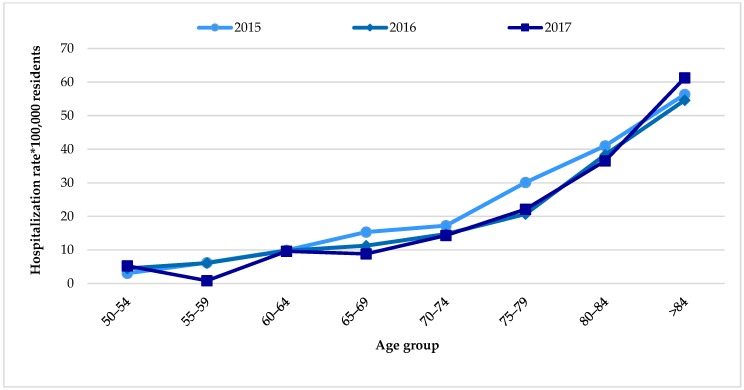
Herpes Zoster (HZ) hospitalization rate per 100,000 residents stratified by age group; years 2015–2017.

**Table 1 vaccines-08-00020-t001:** Prevalence of chronic diseases at 6 months before hospitalization in HZ cohort and non-HZ cohort; period 2015–2017.

Risk Factors	HZ Cohort	Non-HZ Cohort	Odds Ratio	*p*-Value
	(%, 95% CI)	(%, 95% CI)	(OR, 95% CI)	
Transplant	1.83 (0.57; 3.09)	0.34 (0.12; 0.57)	5.41 (2.08; 14.11)	<0.001
Renal failure	10.07 (7.25; 12.89)	8.89 (7.80; 9.98)	1.15 (0.82; 1.61)	0.425
HIV/AIDS	1.37 (0.28; 2.46)	0.04 (−0.04; 0.11)	36.49 (4.38; 303.81)	<0.001
Cancer	18.99 (15.32; 22.67)	21.36 (19.79; 22.93)	0.86 (0.67; 1.12)	0.261
Diabetes	16.48 (13.00; 19.95)	19.45 (17.94; 20.97)	0.82 (0.62; 1.07)	0.142
Non-complicated diabetes	13.5 (10.30; 16.71)	14.45 (13.11; 15.80)	0.92 (0.69; 1.24)	0.598
Complicated diabetes	2.97 (1.38; 4.57)	4.96 (4.13; 5.79)	0.59 (0.33; 1.05)	0.069
Cardiovascular diseases	68.19 (63.83; 72.56)	68.69 (66.91; 70.46)	0.98 (0.79; 1.22)	0.836
Cardiovasc without IPT	37.76 (33.21; 42.30)	35.81 (33.98. 37.65)	1.09 (0.88; 1.34)	0.433
Cerebral vasculopathy	12.1 (9.07; 15.19)	11.9 (10.62; 13.10)	1.03 (0.75; 1.40)	0.873
Hypertension	60.9 (56.29; 65.45)	62.05 (60.19; 63.91)	0.95 (0.77; 1.17)	0.638
Ischemic heart disease	15.56 (12.16; 18.96)	15.22 (13.84; 16.59)	1.03 (0.78; 1.36)	0.854
Valvular heart disease	4.35 (2.44; 6.26)	2.90 (2.26; 3.54)	1.52 (0.91; 2.54)	0.106
Arrhythmic myocardiop.	15.33 (11.95; 18.71)	14.87 (13.51; 16.24)	1.04 (0.78; 1.37)	0.804
Non-arrhythmic myocardiop.	10.98 (8.05; 13.92)	9.38 (8.27; 10.50)	1.19 (0.86; 1.65)	0.294
Heart failure	10.98 (8.05; 13.91)	9.15 (8.05; 10.26)	1.23 (0.88; 1.70)	0.226
Arterial vasculopathy	3.89 (2.08; 5.70)	5.45 (4.58; 6.32)	0.71 (0.42; 1.17)	0.174
Venous vasculopathy	0.23 (−0.22; 0.68)	0.72 (0.40; 1.05)	0.31 (0.04; 2.35)	0.234
Bronchopneumopathy *	14.19 (10.92; 17.46)	13.35 (12.05; 14.65)	1.07 (0.80; 1.44)	0.634
Gastrointestinal diseases	21.97 (18.09; 25.85)	20.25 (18.71. 21.79)	1.11 (0.87; 1.42)	0.411
Gastrointestinal without GERD	3.66 (1.90; 5.42)	3.78 (3.05; 4.51)	0.97 (0.57; 1.66)	0.907
Chronic Hepatitis	2.06 (0.73; 3.39)	1.41 (0.96; 1.86)	1.47 (0.70; 3.07)	0.304
Chronic pancreatitis	0.23 (−0.22; 0.68)	0.31 (0.09; 0.52)	0.75 (6.01; 0.09)	0.785
Liver cirrhosis	1.14 (0.15; 2.14)	1.68 (1.19; 2.17)	0.68 (0.27; 1.72)	0.410
Ulcerative colitis and Crohn	0.23 (−0.22; 0.68)	1.11 (0.71; 1.51)	0.21 (0.03; 1.51)	0.085
GERD	19.91 (16.16; 23.65)	17.81 (16.35; 19.28)	1.15 (0.89; 1.48)	0.293
Neuropathy	7.32 (4.88; 9.77)	8.01 (6.97; 9.05)	0.91 (0.62; 1.34)	0.623
Autoimmune diseases	4.12 (2.26; 5.98)	2.33 (1.75; 2.90)	1.80 (1.06; 3.08)	0.029
Endocrine and metabolic disorders	22.2 (18.30; 26.09)	21.51 (19.94; 23.08)	1.04 (0.82; 1.33)	0.747
Rare diseases	1.14 (0.15; 2.14)	0.31 (0.09; 0.52)	3.78 (1.23; 11.61)	0.013
Psychiatric diseases	1.37 (0.28; 2.46)	2.14 (1.58; 2.69)	0.64 (0.27; 1.49)	0.295
No risk factor	17.62 (14.05; 21.19)	16.44 (15.02; 17.86)	1.09 (0.83; 1.42)	0.539

CI = confidence interval; OR = odds ratio; GERD = gastroesophageal reflux disease; * including asthma.

**Table 2 vaccines-08-00020-t002:** Cumulative incidence rate of chronic diseases during the 6 months after hospitalization in the HZ cohort and non-HZ cohort; period 2015–2017.

Risk Factors	HZ Cohort	Non-HZ Cohort	Relative Risk	*p*-Value
	(%, 95% CI)	(%, 95% CI)	(RR, 95% CI)	
Transplant	0.0 (0.0; 0.0)	0.05 (−0.05; 0.16)	-	0.653
Renal failure	3.07 (1.20; 4.94)	2.53 (1.77; 3.30)	1.21 (0.61; 2.39)	0.592
HIV/AIDS	0.27 (−0.26; 0.80)	0.0 (0.0; 0.0)	-	0.025
Cancer	2.04 (0.43; 3.66)	3.54 (2.54; 4.55)	0.58 (0.25; 1.34)	0.203
Diabetes	2.61 (0.83; 4.40)	1.16 (0.61; 1.71)	2.26 (0.98; 5.18)	0.054
Non-complicated diabetes	2.52 (0.80; 4.25)	1.29 (0.73; 1.85)	1.96 (0.87; 4.40)	0.107
Complicated diabetes	0.84 (−0.11; 1.78)	1.51 (0.94; 2.09)	0.55 (0.17; 1.82)	0.326
Cardiovascular diseases	10.31 (4.26; 16.36)	16.83 (13.55; 20.12)	0.61 (0.33; 1.14)	0.161
Cardiovasc without IPT	11.17 (6.86; 15.47)	13.81 (11.68; 15.93)	0.81 (0.53; 1.22)	0.371
Cerebral vasculopathy	6.13 (3.46; 8.80)	4.14 (3.14; 5.15)	1.48 (0.90; 2.43)	0.144
Hypertension	5.30 (1.48; 9.13)	9.10 (6.89; 11.32)	0.58 (0.27; 1.25)	0.184
Ischemic heart disease	1.27 (0.03; 2.51)	2.23 (1.49; 2.97)	0.57 (0.20; 1.59)	0.283
Valvular heart disease	0.0 (0.0; 0.0)	1.14 (0.65; 1.64)	-	0.043
Arrhythmic myocardiop.	1.32 (0.04; 2.60)	4.06 (3.05; 5.07)	0.32 (0.12; 0.89)	0.023
Non-arrhythmic myocardiop.	4.31 (2.10; 6.52)	2.94 (2.11; 3.77)	1.47 (0.82; 2.63)	0.216
Heart failure	1.83 (0.38; 3.29)	3.25 (2.39; 4.12)	0.56 (0.24; 1.30)	0.182
Arterial vasculopathy	0.28 (−0.27; 0.83)	1.70 (1.09; 2.32)	0.17 (0.02; 1.21)	0.044
Venous vasculopathy	0.0 (0.0; 0.0)	0.0 (0.0; 0.0)	-	-
Bronchopneumopathy *	1.63 (0.21; 3.05)	2.50 (1.72; 3.27)	0.65 (0.26; 1.64)	0.37
Gastrointestinal diseases	7.04 (3.99; 10.09)	3.62 (2.64; 4.61)	1.94 (1.16; 3.24)	0.015
Gastrointestinal without GERD	2.26 (0.71; 3.81)	0.74 (0.34; 1.15)	3.03 (1.27; 7.27)	0.01
Chronic Hepatitis	1.10 (0.028; 2.17)	0.39 (0.10; 0.68)	2.83 (0.83; 9.61)	0.084
Chronic pancreatitis	0.0 (0.0; 0.0)	0.11 (−0.04; 0.26)	-	0.521
Liver cirrhosis	0.82 (−0.10; 1.73)	0.28 (0.04; 0.52)	2.94 (0.71; 12.25)	0.123
Ulcerative colitis and Crohn	0.53 (−0.20; 1.27)	0.28 (0.04; 0.52)	1.93 (0.38; 9.89)	0.426
GERD	5.54 (2.90; 8.17)	3.27 (2.35; 4.19)	1.70 (0.98; 2.95)	0.072
Neuropathy	2.62 (0.93; 4.32)	2.66 (1.88; 3.43)	0.99 (0.49; 2.00)	0.972
Autoimmune diseases	1.40 (0.18; 2.62)	0.22 (0.01; 0.44)	6.25 (1.69; 23.15)	0.002
Endocrine and metabolic disorders	3.65 (1.43; 5.87)	4.15 (3.09; 5.22)	0.88 (0.45; 1.70)	0.713
Rare diseases	0.0 (0.0; 0.0)	0.0 (0.0; 0.0)	-	-
Psychiatric diseases	0.80 (−0.10; 1.71)	0.39 (0.10; 0.69)	2.04 (0.53; 7.85)	0.293

CI = confidence interval; RR = relative risk; GERD = gastroesophageal reflux disease; * including asthma.

**Table 3 vaccines-08-00020-t003:** Cumulative incidence rate of chronic diseases before and after hospitalization period in the HZ cohort.

Risk Factors	Pre-H	Post-H	Post vs. Pre Relative Risk (RR, 95% CI)	*p*-Value
	Incidence	Incidence		
	(%, 95%CI)	(%, 95%CI)		
Transplant	0.27 (−0.26; 0.80)	-	-	0.501
Renal failure	1.46 (0.19; 2.73)	3.07 (1.20; 4.94)	2.09 (0.72; 6.79)	0.179
HIV/AIDS	0.27 (−0.26; 0.79)	0.27 (−0.26; 0.80)	1.01 (0.03; 39.42)	0.995
Cancer	1.96 (0.41; 3.51)	2.04 (0.43; 3.66)	1.04 (0.32; 3.41)	0.946
Diabetes	0.95 (−0.12; 2.02)	2.61 (0.83; 4.40)	2.75 (0.75; 12.83)	0.132
Not-complicated diabetes	0.61 (−0.23; 1.46)	2.52 (0.80; 4.25)	4.11 (0.95; 28.36)	0.059
Complicated diabetes	0.82 (−0.10; 1.73)	0.84 (−0.11; 1.78)	1.03 (0.17; 5.97)	0.98
Cardiovascular diseases	6.61 (2.18; 11.04)	10.31 (4.26; 16.36)	1.56 (0.60; 4.13)	0.358
Cardiovasc. without IPT	2.09 (0.28; 3.91)	11.17 (6.86; 15.47)	5.34 (2.13; 15.77)	<0.001
Cerebral vasculopathy	0.60 (−0.23; 1.42)	6.13 (3.46; 8.80)	10.3 (2.78; 65.47)	<0.001
Hypertension	6.12 (2.25; 10)	5.30 (1.48; 9.13)	0.87 (0.31; 2.37)	0.786
Ischemic heart disease	-	1.27 (0.03; 2.51)	-	-
Valvular heart disease	0.28 (−0.26; 0.82)	-	-	-
Arrhythmic myocardiop.	0.62 (−0.24; 1.48)	1.32 (0.03; 2.60)	2.13 (0.38; 16.59)	0.412
Non-arrhythmic myocardiop.	0.59 (−0.23; 1.41)	4.31 (2.10; 6.52)	7.26 (1.88; 47.16)	0.002
Heart failure	1.18 (0.03; 2.34)	1.83 (0.38; 3.29)	1.55 (0.43; 6.23)	0.515
Arterial vasculopathy	1.10 (0.03; 2.18)	0.28 (−0.27; 0.83)	0.26 (0.01; 2.03)	0.226
Venous vasculopathy	-	-	-	-
Bronchopneumopathy *	1.24 (0.03; 2.44)	1.63 (0.21; 3.04)	1.32 (0.33; 5.51)	0.698
Gastrointestinal diseases	3.41 (1.33; 5.49)	7.04 (3.99; 10.09)	2.06 (0.97; 4.62)	0.062
Gastrointestinal without GERD	-	2.26 (0.71; 3.81)	-	-
Chronic Hepatitis	0.27 (−0.26; 0.80)	1.10 (0.03; 2.17)	4.07 (0.51; 100.6)	0.212
Chronic pancreatitis	-	-	-	-
Liver cirrhosis	0.27 (−0.26; 0.79)	0.82 (−0.10; 1.73)	3.05 (0.33; 80.28)	0.367
Ulcerative colitis and Crohn	-	0.53 (−0.20; 1.27)	-	-
GERD	2.33 (0.63; 4.04)	5.54 (2.90; 8.17)	2.37 (0.99; 6.17)	0.052
Neuropathy	0.56 (−0.21; 1.34)	2.62 (0.93; 4.32)	4.67 (1.11; 31.72)	0.033
Autoimmune diseases	0.28 (−0.27; 0.82)	1.40 (0.18; 1.62)	5.04 (0.70; 120)	0.123
Endocrine and metabolic disorders	4.42 (2.07; 6.77)	3.65 (1.43; 5.87)	0.83 (0.35; 1.9)	0.657
Rare diseases	-	-	-	-
Psychiatric diseases	-	0.80 (−0.10; 1.71)	-	-

CI = confidence interval; RR = relative risk; GERD = gastroesophageal reflux disease; * including asthma.

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
