# Peer review of "The Unknown Health Burden of Herpes Zoster Hospitalizations: The Effect on Chronic Disease Course in Adult Patients ≥50 Years"

_vaccines, 2020, doi:10.3390/vaccines8010020_

Round 1

Reviewer 1 Report

Currently the Glaxo Company is distributing an effective vaccine to prevent herpes zoster (Shingrix). Therefore there is renewed interest in the costs associated with herpes zoster and whether these costs can be reduced by vaccination with Shingrix. The authors of the current manuscript performed a retrospective cohort study to assess patients with severe herpes zoster in one region of Genoa, Italy.

One of the most important results was the finding that the herpes zoster cohort had a 10-fold higher risk for cerebral vasculopathy and a 5-fold higher risk for cardiovascular disease. Thus, this report confirms prior reports but equally importantly fails to find worsening of several other chronic illnesses in patients with severe herpes zoster. Suggestions for improvement are lister below.

Materials Section, Data sources, line 81. Not all readers of this article will be familiar with the health care delivery system in the Liguria Region. Please write a few sentences to clarify if these data represent the entire population of Liguria, including all socioeconomic classes. In other words, are there private hospitals in Liguria which would not have been included in the data warehouse? Results, lines 138-145. These positive results may be the most important results in the report. The results are only given in percentages. Just for data in this paragraph, provide the actual number of patients in the herpes zoster group and the control group in each of the three categories (cardiovascular, cerebral vasculopathy, and neuropathy). For example, 11.17% = how many patients? And 2.09% = how many controls?

Results, lines 146-148. Since these results are not significant, would delete sentence.

4.Discussion, lines 197-201. The current opinion is that herpes zoster of the trigeminal ganglion allows virus to travel in afferent fibers to the cerebral artery, where the infection slowly leads to inflammation and thrombosis and subsequent ischemic stroke. There is no rupture of artery. Please add the following 2 references at this location, in order to add clarity to the manuscript. (4A) Lin HC, Chien CW, Ho JD. Herpes zoster ophthalmicus and the risk of stroke: a population-based follow-up study. Neurology 74:792-7, 2010. (4B) Grose C, Adams H. Reassessing the link between herpes zoster ophthalmicus and stroke. Expert Review of Anti-infective therapy 12:527-30, 2014.

5.Addition to Discussion. Negative data can be as important as positive data. The authors have clearly shown that many chronic diseases did not worsen in the 6-months after herpes zoster. Please write a new paragraph to discuss negative findings. The suggestion is also made that the authors delete the paragraph between lines 219-224. For example, the authors do not appear to have found an increase in asthma. See Comment 8.

6.References. Add the 2 new references mentioned in Comment 4a and 4b.

7.Tables 1, 2, and 3. There are many numbers in these Tables. To make the important data easier to find, please BOLD all the significant p values in the “p-value” column in all 3 Tables (or put an asterisk next to them).

8.Question about asthma. Asthma is mentioned in the Introduction (line 49). But asthma is not mentioned in any Table. Is asthma a part of bronco-pneumopathy? If yes, please add the word asthma under the word bronco-pneumopathy, at least in Table 1. See comment 5.

Author Response

Response to Reviewer 1 Comments:

Point 1 Materials Section, Data sources, line 81. Not all readers of this article will be familiar with the health care delivery system in the Liguria Region. Please write a few sentences to clarify if these data represent the entire population of Liguria, including all socioeconomic classes. In other words, are there private hospitals in Liguria which would not have been included in the data warehouse?

Response 1: Thank you for your valuable comment and accordingly we have specified this aspect in the material and methods section (lines 69-71). However, in the first version of the manuscript, we have already reported this aspect in the discussion section (lines 228-231).

In this study, the subjects residing in Liguria Region with and without a hospitalization for HZ from January 2015 to December 2017 were enrolled including public and private hospitals. In Italy the hospital access is free of charge.

The administrative healthcare data (AHD) “Data warehouse” is a regional service that collects Hospital Discharge Records (HDRs), the flow of outpatient visits and pharmaceutical of all subjects who access to the Regional Health Care System.

Moreover, as described by Trucchi et al. the Liguria Chronic Condition Data Warehouse (CCDWH) platform, used to evaluate the chronic diseases, gathered from multiple Medicare data sources (HDRs, pharmaceutics, medical fee exemptions and outpatient visits) within a specified period by means of a predefined algorithm based on the codes assigned to specific diagnoses and procedures. Through a record-linkage system based on a civil registry database, residents’ histories of healthcare events are constructed in order to depict the chronic condition of each patient. All data are archived in a relational database by means of big-data logic.

Point 2: Results, lines 138-145. These positive results may be the most important results in the report. The results are only given in percentages. Just for data in this paragraph, provide the actual number of patients in the herpes zoster group and the control group in each of the three categories (cardiovascular, cerebral vasculopathy, and neuropathy). For example, 11.17% = how many patients? And 2.09% = how many controls?

Response 2: Thank you for your valuable suggestion and we modified the result section as suggested.

Point 3: Results, lines 146-148. Since these results are not significant, would delete sentence.

Response 3: We appreciate your suggestion, but we consider this aspect relevant; it could be explored in further analysis with a larger sample sizes giving more reliable results with greater precision and power.

Point 4: Discussion, lines 197-201. The current opinion is that herpes zoster of the trigeminal ganglion allows virus to travel in afferent fibers to the cerebral artery, where the infection slowly leads to inflammation and thrombosis and subsequent ischemic stroke. There is no rupture of artery.

Please add the following 2 references at this location, in order to add clarity to the manuscript. (4A) Lin HC, Chien CW, Ho JD. Herpes zoster ophthalmicus and the risk of stroke: a population-based follow-up study. Neurology 74:792-7, 2010. (4B) Grose C, Adams H. Reassessing the link between herpes zoster ophthalmicus and stroke. Expert Review of Anti-infective therapy 12:527-30, 2014.

Response 4: We agree with your suggestion and we have modified the text as requested.

Furthermore, we have modified the reference number 19, because it was not correct. The correct reference is “Zhang, Y.; Luo, G.; Huang, Y.; Yu, Q.; Wang, L.; Li, K. Risk of Stroke/Transient Ischemic Attack or Myocardial Infarction with Herpes Zoster: A Systematic Review and Meta-Analysis. J Stroke Cerebrovasc Dis. 2017, 26(8):1807-1816”, which we have replaced in the previous reference.

Finally, we added the two references suggested and modified the number of the following references in the text.

Point 5: Addition to Discussion. Negative data can be as important as positive data. The authors have clearly shown that many chronic diseases did not worsen in the 6-months after herpes zoster. Please write a new paragraph to discuss negative findings. The suggestion is also made that the authors delete the paragraph between lines 219-224. For example, the authors do not appear to have found an increase in asthma. See Comment 8.

Response 5: We appreciated your suggestion and we added a brief paragraph in discussion section (lines 238-242).

Furthermore, the sentence between lines 219-224 (lines 223-224 in revised version) was summarized.

Point 6: Add the 2 new references mentioned in Comment 4a and 4b.

Response 6: As suggested, we added the references mentioned in the previous comment.

Point 7: Tables 1, 2, and 3. There are many numbers in these Tables. To make the important data easier to find, please BOLD all the significant p values in the “p-value” column in all 3 Tables (or put an asterisk next to them).

Response 7: Thank you for your valuable suggestion and we adapted the tables highlighting in BOLD the significant p values.

Point 8: Question about asthma. Asthma is mentioned in the Introduction (line 49). But asthma is not mentioned in any Table. Is asthma a part of bronco-pneumopathy? If yes, please add the word asthma under the word bronco-pneumopathy, at least in Table 1. See comment 5.

Response 8: Asthma is included in broncopneumopathy category and we specified it in the table 1, 2 and 3. Furthermore, considering the Herpes Zoster epidemiology that involve mainly older subjects, it is reasonable to think that asthma is not a frequent disease in this age group.

Reviewer 2 Report

The authors have presented a retrospective cohort study to assess the health burden of severe herpes zoster (HZ) on older adults, including those with chronic diseases, a component of the burden of herpes zoster disease. They found that previously identified autoimmune, immunodeficiency and rare diseases were the main chronic conditions related to hospitalisation of the HZ cohort. Further, the HZ cohort showed a higher frequency of autoimmune and gastrointestinal diseases after hospitalisation than the non-HZ cohort, and higher incidences were found after hospitalisation of cardiac and related events. This study has been well justified, and is an important investigation of a poorly considered burden of a vaccine-preventable disease, HZ. The data collected have been rigorously interrogated, but not over-interpreted, and presented in a clear and understandable manner. Further, the authors should be commended for acknowledging the (very reasonable) limitations of their study in the discussion. Further, this study may have important public health implications for those elderly and high risk patients, and represents an important contribution to the literature. I have no recommendations for this study, and believe it should be published as submitted.

Author Response

Dear Reviewer, my colleagues and I are most grateful for your comments of our paper “The unknown health burden of Herpes Zoster hospitalizations: the effect on chronic disease course in adult patients ≥ 50 years”.